# The impact of incomplete data on quantile regression for longitudinal data

**Anneleen Verhasselt** [1]  **Alvaro J. Flórez** [1]  **Geert Molenberghs** [1,2]  **Ingrid Van Keilegom** [3]

## Abstract

We investigate the performance of quantile methods for longitudinal data with missingness. In a simulation study, we compare the performance of the quantile regression using different alternatives for handling missing data and taking the correlation into account. As expected, the non-likelihood-based methods provide biased estimates under the missing at random assumption. On the other hand, an inverse probability weighting approach corrects for biasedness.

## 1. Introduction

In longitudinal studies, the same characteristics of individuals is repeatedly measured over time, allowing them to analyze their changes over time. Quantile regression (QR; Koenker & Bassett, 1978; Koenker, 2005) permits examining the effect of a set of covariates on different quantiles of a response variable. Therefore, it is useful for analyzing this type of data when the distribution of the responses is skewed, the data contain outliers, or when flexibility to the error distribution is relevant.

Motivated by the well-known equivalence in the univariate QR estimator (minimization of the check function) and the maximization of the likelihood based on an asymmetric Laplace (AL) distribution, we consider to estimate the quantiles with correlated data by maximizing the likelihood-based on a multivariate extension of the AL distribution (Kozubowski & Podgórski, 2000; Kotz et al., 2001). Note that the multivariate distribution allows addressing the dependence between the longitudinal observations into account, whereas the classical univariate QR ignores this dependence. (Petrella & Raponi, 2019) showed via simulations that, despite the peaks and non-differentiability problems inherent to the latter distribution, it is possible

*Equal contribution [1]I-BioStat, Universiteit Hasselt, Diepenbeek, Belgium [2]I-BioStat, KU Leuven, Leuven, Belgium [3]ORSTAT, KU Leuven, Leuven, Belgium. Correspondence to: Alvaro J. Flórez <alvaro.florez@uhasselt.be>.

*Presented at the first Workshop on the Art of Learning with Missing Values (Artemiss) hosted by the 37th International Conference on Machine Learning (ICML). Copyright 2020 by the author(s).*

to estimate the model correctly. However, as (Kotz et al., 2001) point out, there are some issues with the multivariate asymmetric Laplace (MAL) distribution that require special treatment and attention. We solve these issues and show that the estimator is asymptotically normal and that standard errors can be computed via a minor modification in the likelihood function.

Missing data occurs when not all scheduled measurements of a subjects outcome are observed. The nature of the missingness mechanism highly influences the performance of statistical techniques that deal with missing data. Therefore, it is important to define the mechanism. There are three main missing data mechanisms (Rubin, 1976). Under missing completely at random (MCAR), missingness does not depend on either the observed or unobserved variables, apart from perhaps covariates. When missingness is independent of the unobserved measurements conditional on the observed ones, the process is called missing at random (MAR). Missing not at random (MNAR) occurs when neither MAR nor MCAR holds.

Under the most common assumption MAR, the full likelihood methods provide valid estimates. On the contrary, non-likelihood estimators, such as the classical QR, can provide biased estimates (Molenberghs et al., 2011). Therefore, in this paper, we compare under simulations different approaches for estimating conditional quantiles for longitudinal data in the presence of missing values. Particularly, we focus on dropouts, where subjects drop out of the study at a certain occasion, and there are no recordings afterward.

## 2. Model and methodology

We focus on estimating the $\tau$-th quantile (with $0 < \tau < 1$) of a response given the covariates. Suppose that $\mathbf{Y}_i = (Y_1, \ldots, Y_n)'$ is an $n$-dimensional response vector for individual $i = 1, \ldots, N$. Consider the multivariate regression model:

$$\mathbf{Y}_i = \mathbf{X}_i \boldsymbol{\beta} + \boldsymbol{\varepsilon},$$

where $\mathbf{X}_i$ is a $(n \times p)$-design matrix of covariates, $\boldsymbol{\beta} = (\beta_1, \ldots, \beta_p)'$ is a vector of regression coefficients, and $\boldsymbol{\varepsilon} = (\varepsilon_{i1}, \ldots, \varepsilon_{in})$ is a vector of error terms. Note that the $\tau$-th conditional quantile of $\mathbf{Y}_i$ is

$$Q_\tau(\mathbf{Y}_i | \mathbf{X}_i) = \mathbf{X}_i' \boldsymbol{\beta} + Q_\tau(\boldsymbol{\varepsilon}_i | \mathbf{X}_i).$$

## 2.1. Quantile regression

When estimating the $\tau$-th conditional quantile of $\mathbf{Y}_i$, we assume that $Q_\tau(\boldsymbol{\varepsilon}_i|\mathbf{X}_i) = \mathbf{0}$. This is commonly assumed in quantile regression, when estimating one specific quantile. With this assumption, the $\tau$-th conditional quantile of $\mathbf{Y}_i$ is given by

$$Q_\tau(\mathbf{Y}_i|\mathbf{X}_i) = \mathbf{X}_i'\boldsymbol{\beta}.$$

This conditional quantile can easily be estimated, given an estimator for $\boldsymbol{\beta}$. (Koenker & Bassett, 1978) proposed the following quantile regression estimator for $\boldsymbol{\beta}$:

$$\hat{\boldsymbol{\beta}} = \arg\min_{\boldsymbol{\beta}} \sum_{i=1}^{N} \sum_{j=1}^{n} \rho_\tau(Y_{ij} - \mathbf{x}_{ij}'\boldsymbol{\beta}), \qquad (1)$$

where $\mathbf{x}_{ij}$ is the $j$th row of $\mathbf{X}_i$, $\rho_\tau(u) = u[\tau - I(u < 0)]$ is the check-loss function used in quantile regression.

The check function $\rho_\tau(\cdot)$ is proportional to the negative log density of the asymmetric Laplace distribution. This connection lets us assume that $Y$ is distributed as an asymmetric Laplace, denote as $Y \sim AL(\mu, \phi, \tau)$, with probability density function given by:

$$f(y|\mu, \phi, \tau) = \frac{\tau(1-\tau)}{\phi} \exp\left[-\rho_\tau\left(\frac{y-\mu}{\phi}\right)\right], \qquad (2)$$

where $\mu$ is a location parameter, $\phi > 0$ is a scale parameter, and $\tau$ plays the role of skewness parameter.

Then, by assuming that $y_{ij} \sim AL\left(\mu_{ij} = \mathbf{x}_{ij}'\boldsymbol{\beta}, \phi, \tau\right)$, $\boldsymbol{\beta}$ is estimated by maximizing the log-likelihood, defined as:

$$\ell(\boldsymbol{\beta}, \phi) \propto -n\log(\phi) - \sum_{i=1}^{N}\sum_{j=1}^{n} \rho_\tau\left(\frac{y_{ij} - \mathbf{x}_{ij}\boldsymbol{\beta}}{\phi}\right),$$

which is equivalent to the minimization of the objective function (1). For more details on univariate QR, we refer to (Koenker, 2005).

## 2.2. Quantile regression for longitudinal data

Firstly, we consider the multivariate asymmetric Laplace distribution (Kozubowski & Podgórski, 2000), $\mathbf{Y}_i \sim \mathrm{MAL}_n\left(\mathbf{X}_i\boldsymbol{\beta}, \boldsymbol{\Delta}\boldsymbol{\xi}, \boldsymbol{\Delta}\boldsymbol{\Sigma}\boldsymbol{\Delta}\right)$, with density:

$$f_{\mathbf{Y}}(\mathbf{y}; \boldsymbol{\theta}) = \frac{2\exp\left[(\mathbf{y} - \mathbf{X}_i\boldsymbol{\beta})'\boldsymbol{\Delta}^{-1}\boldsymbol{\Sigma}^{-1}\boldsymbol{\xi}\right]}{(2\pi)^{n/2}|\boldsymbol{\Delta}\boldsymbol{\Sigma}\boldsymbol{\Delta}|^{1/2}}\left(\frac{m_i}{2+d}\right)^{\nu/2} \times$$
$$\times K_\nu\left[\sqrt{(2+d)m_i}\right],$$

where $\boldsymbol{\Delta}\boldsymbol{\xi}$ is the scale (or skewness) parameter vector and $\boldsymbol{\Sigma} = \boldsymbol{\Lambda}\boldsymbol{\Psi}\boldsymbol{\Lambda}$ a positive definite matrix. Furthermore, $\boldsymbol{\Delta} = \mathrm{diag}(\delta_1, \ldots, \delta_n)$, $\delta_j > 0$ (for $j = 1, \ldots, n$), $\boldsymbol{\xi} = (\xi_1, \ldots, \xi_n)'$, $\xi_j = \frac{1-2\tau}{\tau(1-\tau)}$ for $j = 1, \ldots, n$, $\boldsymbol{\Lambda} = \mathrm{diag}(\lambda_1, \ldots, \lambda_n)$, $\lambda_j^2 = \frac{2}{\tau(1-\tau)}$ for $j = 1, \ldots, n$,

and $\boldsymbol{\Psi}$ is a correlation matrix. Further $m_i = (\mathbf{y} - \mathbf{X}_i\boldsymbol{\beta})'(\boldsymbol{\Delta}\boldsymbol{\Sigma}\boldsymbol{\Delta})^{-1}(\mathbf{y} - \mathbf{X}_i\boldsymbol{\beta})$, $d = \boldsymbol{\xi}'\boldsymbol{\Sigma}\boldsymbol{\xi}$, and $K_\nu$ is the modified Bessel function of the third kind with index parameter $\nu = (2-n)/2$.

We consider a maximum likelihood estimator (MLE). However, the log-likelihood function diverges to infinity when $\mathbf{y}$ tends to $\mathbf{X}_i\boldsymbol{\beta}$ leading to serious computational issues. This is because the Bessel function $K_\nu(u)$ is proportional to $u^{-\nu}$ for $u$ close to zero (see the Appendix of (Kozubowski & Podgórski, 2000)). Therefore, the Bessel function $K_\nu(\sqrt{(2+d)m_i})$ in (3) is slightly modified by $K_\nu(\sqrt{(2+d)m_i} + \epsilon)$ for some small $\epsilon > 0$.

Since the MLE can be computationally intensive for estimating high-dimensional data, we consider pseudo-likelihood approaches (Molenberghs et al., 2011). Particularly, a pairwise estimator (PWE). Let $S$ be the set of all $n!/[2!(n-2)!]$ vectors of length $n$ consisting of zeros and ones, with each vector having exactly two non-zero entries. Denote by $\mathbf{Y}_i^{(s)}$ the subvector of $\mathbf{Y}_i$ corresponding to the components of $s$ that are non-zero. The associated joint density is $f_{\mathbf{Y}^{(s)}}(\mathbf{y}^{(s)}; \boldsymbol{\theta}^s)$. Then, the pairwise estimator maximizes the pseudo-log-likelihood:

$$p\ell(\boldsymbol{\theta}) = \sum_{i=1}^{N}\sum_{s\in S} \varphi_s \log f_{\mathbf{Y}^{(s)}}(\mathbf{y}_i^{(s)}; \boldsymbol{\theta}^{(s)}),$$

where $\varphi = \{\varphi_s | s \in S\}$. Note that the classical log-likelihood function is found setting $\varphi_s = 1$ if s is the vector consisting solely of ones, and zero otherwise.

# 3. Quantile regression with missing data

For non-fully-likelihood-based methods, we contemplate inverse probability weighting (IPW) methods (Robins et al., 1994; 1995). Here, the contributions are weighted by the inverse probability of being observed. For instance, for univariate QR, The IPW estimator of $\boldsymbol{\beta}$ is:

$$\hat{\boldsymbol{\beta}} = \arg\min_{\boldsymbol{\beta}} \sum_{i=1}^{n}\sum_{j=1}^{n_i} \frac{R_{ij}}{\pi_{ij}}\rho_\tau(Y_{ij} - \mathbf{X}_{ij}'\boldsymbol{\beta}),$$

where $R_{ij} = 1$ if $y_{ij}$ is observed, $R_{ij} = 0$ otherwise, and $\pi_{ij}$ is the probability of $y_{ij}$ being observed.

For the PWE, we maximize following weighted pseudo-likelihood function:

$$p\ell(\boldsymbol{\theta}) = \sum_{i=1}^{N}\sum_{s\in S} \frac{R_i^{(s)}}{\pi_i^{(s)}} \log f_{\mathbf{Y}^{(s)}}(\mathbf{y}_i^{(s)}; \boldsymbol{\theta}^{(s)}),$$

where $R_i^s = 1$ if the pair $y_i^s$ is observed, $R_i^{(s)} = 0$ otherwise, and $\pi_i^s$ is the probability of the pair $y_i^s$ being observed.

The probabilities $\pi_{ij}$ $(j = 2, \ldots, n_i)$ are obtained as follows (assuming that the first time point is always observed):

- if the subject drops out at occasion $j$: $\pi_{ij} = p_{ij} \prod_{l=2}^{j-1} (1 - p_{il})$

- if the subject does not drop out at occasion $j$: $\pi_{ij} = \prod_{l=2}^{j} (1 - p_{il})$,

with $p_{il} = P(D_i = l | D_i \geq l, \mathbf{Y}_{i\bar{l}}, \mathbf{X}_i)$ (the probability of dropping out at occasion $l$ given the subject is still in the study) where $\mathbf{Y}_{i\bar{l}} = (Y_{i1}, \ldots, Y_{i(l-1)})'$ is the outcome history. In practice, the probabilities $p_{il}$ are unknown and need to be estimated, for example by assuming a logistic regression model using the outcome $\mathbf{Y}_{i\bar{l}} = (Y_{i1}, \ldots, Y_{i(l-1)})'$ and covariates $\mathbf{X}_i$ as regressors.

# 4. Simulation study

## 4.1. Settings

The setting resembles longitudinal data with three measurements per subjects with missing observations at the third time. The data-generating model is:

$$Y_{ij} = \beta_0 + x_j\beta_1 + x_j z_i \beta_2 + (\gamma_0 + x_j\gamma_1 + x_j z_i \gamma_2)\varepsilon_{ij},$$

for $i = 1, \ldots, N$, $j = 1, 2, 3$, and with $x_j = (j - 1)/3$ indicating the measurement time, and $z_i$ representing a Bernoulli variable with sucess probability 0.5. We assume that $\boldsymbol{\varepsilon}_i = (\varepsilon_{i1}, \varepsilon_{i2}, \varepsilon_{i3})' \sim N(\mathbf{0}, \boldsymbol{\Sigma})$, with:

$$\boldsymbol{\Sigma} = \begin{pmatrix} 1 & 0.2 & 0.5 \\ 0.2 & 1 & 0.7 \\ 0.5 & 0.7 & 1 \end{pmatrix}.$$

We contemplate missing observations on $j = 3$, in which the probability of missingness is determined by

$$P(R_{i3} = 0) = \frac{\exp(\alpha_0 + \alpha_1 Y_{i2} + \alpha_2 z_i)}{1 + \exp(\alpha_0 + \alpha_1 Y_{i2} + \alpha_2 z_i)},$$

where $R_{ij} = 0$ if $y_{ij}$ is missing, and $R_{ij} = 1$ otherwise.

For the simulations, we set $\beta = (\beta_0, \beta_1, \beta_2)' = (4, 2, 1)'$, $\boldsymbol{\gamma} = (1, 0.414, 0)$, and a sample size of $N = 200$.

## 4.2. Estimators

For estimating the quantiles, we implement the following estimators:

- UQR: the univariate quantile regression estimator. In case of missing data, we consider the available cases (AC), and inverse probability weighting (IPW).

- MLE: the maximum likelihood estimator based on the MAL distribution. In case of missigness, we contemplate the available cases (AC).

- PWE: the pairwise estimator based on the MAL distribution. In case of missing data, we consider the complete pairs (CP), and inverse probability weighting (IPW).

Note that the MLE is likelihood-based. Consequently, a bias-correction method is not required.

## 4.3. Results

Each scenario is simulated $M = 500$ datasets. Furthermore, we analyze the estimators for estimating quantiles $\tau = \{0.25, 0.5, 0.9\}$. On each regression coefficient separately, we compute the relative bias (RB) and the squared root of the mean square error (SQMSE). The former is defined as:

$$RB(\widehat{\beta}_j^\tau) = \frac{1}{M} \sum_{k=1}^{M} \frac{\widehat{\beta}_{jk}^\tau - \beta_j}{\beta_j},$$

and the latter as:

$$RMSE(\widehat{\beta}_j^\tau) = \sqrt{\frac{1}{M} \sum_{k=1}^{M} (\widehat{\beta}_{jk}^\tau - \beta_j)^2}$$

Table 1 displays the RE and SQMSE of the estimators for each parameter and different values of $\tau$. Considering the full data, MLE and PWE are unbiased and more efficient than UQR. This result is expected because these two estimators take into account the association of the data. Regarding missing data, the UQR and PWE require an IPW approach for bias-correction. However, although there is a noticeable reduction of the bias, the variability is large. On the other hand, the MLE still provides unbiased estimates under MAR.

Regarding computation time, the MLE required, on average, 15 seconds fitting the model. On the other hand, the PWE took roughly 2 seconds. For higher dimensions, we expect that the MLE is computationally too intensive or even untreatable.

# 5. Final remarks

We considered a quantile regression model for longitudinal data with missingness in the response. Using simulations, we investigated the impact of correlation and missing data on estimating the regression coefficient using different estimators. The MLE based on the MAL distribution takes into account the dependence structured of the data, and therefore, is more efficient. However, it is computationally more intensive. For this reason, a pairwise estimator is also proposed. Since this is a non-likelihood-based method, an inverse probability weighting approach is required for bias-correction under missingness.

Table 1. Relative bias (in percentage) and the squared root of the mean square error of the univariate quantile regression (UQR), maximum likelihood estimator (MLE), pairwise estimator (PWE) with full and missing data and different values of $\tau$.

| | | Relative bias (%) | | | | | | | |
| | | UQR | | | MLE | | PWE | | |
| $\tau$ | parm | full | AC | IPW | full | AC | full | CP | IPW |
|---|---|---|---|---|---|---|---|---|---|
| 0.25 | $\beta_0$ | 0.04 | 1.76 | -0.19 | -0.61 | -0.48 | -0.80 | -2.87 | -0.72 |
| | $\beta_1$ | 0.00 | -21.38 | 1.55 | -1.08 | -3.05 | -0.24 | -15.19 | -0.44 |
| | $\beta_2$ | -1.50 | -8.03 | 3.43 | -0.58 | 0.46 | -1.24 | -8.91 | 0.11 |
| 0.5 | $\beta_0$ | -0.03 | 1.60 | -0.30 | -0.09 | -0.07 | -0.08 | -1.83 | 0.13 |
| | $\beta_1$ | 0.04 | -21.25 | 2.05 | 0.26 | -0.09 | 0.16 | -15.54 | -0.20 |
| | $\beta_2$ | -1.35 | -6.50 | 6.30 | -0.91 | -0.04 | -1.06 | -7.92 | 1.13 |
| 0.9 | $\beta_0$ | -0.12 | 1.52 | 0.40 | 1.48 | 1.46 | 1.43 | 0.36 | 2.14 |
| | $\beta_1$ | 0.23 | -21.00 | -3.96 | 4.63 | 2.68 | 1.18 | -16.81 | -4.44 |
| | $\beta_2$ | -0.57 | -3.98 | 0.18 | -1.31 | -0.58 | -0.69 | -4.08 | -3.02 |
| | | Sqrt. mean square error | | | | | | | |
| | | UQR | | | MLE | | PWE | | |
| $\tau$ | parm | full | AC | IPW | full | AC | full | CP | IPW |
| 0.25 | $\beta_0$ | 0.08 | 0.10 | 0.09 | 0.08 | 0.08 | 0.08 | 0.13 | 0.10 |
| | $\beta_1$ | 0.19 | 0.44 | 0.38 | 0.15 | 0.19 | 0.16 | 0.33 | 0.22 |
| | $\beta_2$ | 0.27 | 0.32 | 0.61 | 0.19 | 0.25 | 0.21 | 0.28 | 0.33 |
| 0.5 | $\beta_0$ | 0.07 | 0.10 | 0.08 | 0.07 | 0.07 | 0.06 | 0.10 | 0.10 |
| | $\beta_1$ | 0.17 | 0.47 | 0.43 | 0.13 | 0.17 | 0.13 | 0.35 | 0.24 |
| | $\beta_2$ | 0.25 | 0.29 | 0.74 | 0.20 | 0.24 | 0.20 | 0.25 | 0.41 |
| 0.9 | $\beta_0$ | 0.11 | 0.14 | 0.12 | 0.12 | 0.14 | 0.12 | 0.11 | 0.28 |
| | $\beta_1$ | 0.27 | 0.60 | 0.63 | 0.22 | 0.27 | 0.21 | 0.50 | 0.60 |
| | $\beta_2$ | 0.35 | 0.40 | 0.92 | 0.20 | 0.26 | 0.25 | 0.28 | 0.72 |

**full:** full data, **AC:** available cases, **CP:** complete pairs, **IPW:** inverse probability weighting

Although the IPW approach corrects for biasedness, it can be inefficient. Therefore, an augmented inverse probability weighting (AIPW) can be considered. Here, IPW is expanded by a term with contributions from individuals with missing data into the estimating equation (Robins et al., 1994; 1995). We expect that his class of estimators improve efficiency. Furthermore, the statistical and computational performance of these estimators for high-dimensional data with a wide range of dependence structures should be evaluated.

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
