# OpenReview forum: "The impact of incomplete data on quantile regression for longitudinal data"
_ICML.cc/2020/Workshop/Artemiss — ICML Artemiss 2020_

### Official Review · AnonReviewer1 · 2020-06-23
**Interesting, but slightly below the threshold due to lack of a clear motivation and experiments.**

**Rating:** 5
**Confidence:** 3

**Review:**

Summary:
This paper proposes to assess multiple methods for quantile regression (QR) with missing data. For quantile regression, the paper studies two methods: 1) multivariate asymmetric Laplace (AL) distributions, and 2) univariate AL combined via copulas. For non-likelihood-based QR estimators, missingness was handled with multiple imputations (MI) as well as inverse probability weighting (IPW). In a simulation study, the paper finds that the non-likelihood-based approaches are biased in the MAR setting, whereas all methods work fine under MCAR. For non-likelihood-based approaches, MI was more efficient and less biased and IPW, however requiring more computational ressources.

Assessment:
Overall, I believe the paper suffers a bit from low clarity and insufficient motivation. I had to read the paper several times until I grasped what the rough message of the paper should be. In my view,
the paper could benefit from the following points:
1. a clearer motivation: why should we be doing QR, why is this relevant, what are applications. What is the current literatur    missing?
2. a clearer distinction between existing work and original contributions
3. experimental results (I know workshop submission can also be preliminary, but showing simulation results I think should be an absolut minimum)
Furthermore, I could imagine that a more statistical venue could be more appropriate and appreciative for this work.

Minor points:
- The abbreviation AL should be introduced.
- I found the notation a bit overloaded w.r.t. the usage of N on the individual level and n as the dimensionality of each individuals data.

---

### Official Review · AnonReviewer2 · 2020-06-23
**Relevant but focus could be shifted more to missing data**

**Confidence:** 3
**Rating:** 7

**Review:**

The paper deals with MAR missing values in longitudinal outcome variables due to drop-out and how MI and IPW can be used to reduce bias when such data is analysed with quantile regression.

I find this an interesting and relevant research question, also because inference about quantiles is often less robust and hence appropriate handling of missing values maybe even more crucial than when inference is made about a mean.

The authors consider multiple imputation to impute missing values before analysis, but it is not clear how this multiple imputation is applied (joint model, FCS?) The more common FCS is not well suited for imputation in longitudinal settings, at least not without tweaking.

The paper is entitled "The impact of incomplete data on ...", however, most of the paper deals with the technical details of quantile regression. From the title, I would expect a comparison of the performance in complete vs incomplete data. Since comparison is done between two methods to handle the missing data, maybe more focus on the description of those methods would be appropriate.

I would have liked to get more information on the simulation study, the scenarios and the results.
The only result reported is that both methods reduce bias and that MI is more efficient. How much reduction? Enough that we can consider either method to be a solution to the problem? How practical are both methods to use in practice? E.g., the authors mention computational time as an issue (minutes, hours, days?) and can the difference in efficiency be quantified? Maybe the authors can visit some of these questions during their presentation, to balance theoretical and practical information a bit.

---

### Decision · Program_Chairs · 2020-07-02

**Decision:**

Accept

**Comment:**

We are very happy to inform you that your paper has been accepted for the Artemiss workshop. We will contact you soon to inform you about the details concerning the format of your presentation at the workshop, and the camera-ready version deadline. Please take into account the referee's comments to write the camera-ready version.